# Brief Communication: Critical Infrastructure impacts of the 2021 mid-July western European flood event

Elco E. Koks[1,2], Kees C.H. Van Ginkel[3,1*], Margreet J.E. Van Marle[3], Anne Lemnitzer[4]

1. Institute for Environmental Studies, Vrije Universiteit Amsterdam, The Netherlands
2. Environmental Change Institute, University of Oxford, Oxford, United Kingdom
3. Deltares, Delft, The Netherlands
4. University of California, Irvine, United States of America.
* Corresponding author

*Correspondence to*: Elco E. Koks (elco.koks@vu.nl)

**Abstract.** Germany, Belgium and The Netherlands were hit by extreme precipitation and flooding in July 2021. This Brief Communication provides an overview of the impacts to large-scale critical infrastructure systems and how recovery has progressed during the first six months after the event. The results show that Germany and Belgium were particularly affected, with many infrastructure assets severely damaged or completely destroyed. Impacts range from completely destroyed bridges and sewage systems, to severely damaged schools and hospitals. We find that (large-scale) risk assessments, often focused on larger (river) flood events, do not find these local, but severe, impacts due to critical infrastructure failures. This may be the result of limited availability of validation material. As such, this Brief Communication will not only help to better understand how critical infrastructure can be affected by flooding, but can also be used as validation material for future flood risk assessments.

## 1 Introduction

In mid-July 2021, a persistent low-pressure system caused extreme precipitation in parts of the Belgian, German and Dutch catchments of the Meuse and Rhine river. This led to record breaking water levels and severe flooding (Mohr et al., 2022). Comparable heavy precipitation events in this area have never been registered in most of the affected areas before (Kreienkamp et al., 2021). The German states most affected include Rheinland-Pfalz (Rhineland-Palatinate), with damage to the Ahr river valley (Ahrtal), several regions in the National Park "Eifel", as well as the city of Trier. Flooding in Belgium was concentrated in the Vesdre river valley (districts of Pepinster, Ensival, and Verviers), the Meuse river valley (Maaseik, Liege), the Gete river valley (Herk-De-Stad and Halen) and southeast Brussels (Wavre). The Netherlands experienced flooding, mostly concentrated in the southern district of Limburg. In total, at least 220 casualties have been reported, with insured loss estimates of approximately 150-250 million EUR in The Netherlands (Verbond voor Verzekeraars, 2022), ~2.2 billion EUR in Belgium (Assuralia, 2021) and ~8.2 billion EUR (GDV, 2022) in Germany. The event not only caused major damages to residential and commercial structures, but also to critical infrastructure (CI) in particular. Not only vital functions in the first response

were affected (e.g., hospitals, fire departments), but also railways, bridges, and utility networks (e.g., water and electricity supply) were severely damaged, expecting to take months to years to fully rebuild.

CI is often considered to be the backbone of a well-functioning society (Hall et al., 2016), which is particularly eminent during natural hazards and disasters. For instance, failure of electricity or telecommunication services immediately causes disruptions in the day-to-day functioning of people and businesses, including those outside the directly affected area. Despite the (academic) agreement that failure of infrastructure systems may cause (large-scale) societal disruptions (Garschagen and Sandholz, 2018; Hallegatte et al., 2019; Fekete and Sandholz, 2021), empirical evidence on the impacts of extreme weather events on these systems is still limited. This Brief Communication provides an overview of the observed flood impacts to large-scale infrastructure systems during the 2021 mid-July western European flood event, and how reconstruction of these large-scale systems has progressed. Next, we highlight how some of these observations compare to academic modelling approaches. We conclude with suggestions on moving forward in CI risk modelling, based on the lessons learned from this extreme event.

## 2 Critical Infrastructure Impacts

### 2.1 Transport Infrastructure

In Germany, road and railway infrastructure has been severely damaged as documented exemplarily in Figure 1. Cost estimates reach up to 2 billion Euro (MDR, 2021a). More than 130 km of motorways were closed directly after the event, 50 km were still closed two months later, with an estimated repair cost of 100 million Euro (Hauser, 2021). Of the 112 bridges in the flooded 40 km of the Ahr Valley (Rheinland Pfalz), 62 bridges were destroyed, 13 were severely damaged, and only 35 were in operation a month after the flood event (Hochwasser Ahr, 2021). Over 74 km of roads, paths and bridges in the Ahr valley have been (critically) damaged. In some cases, repairs are expected to take months to years (Zeit Online, 2021). For example, major freeway sections, including parts of the A1 motorway were closed until early 2022 (24Rhein, 2022). In addition, about 50,000 cars were damaged, causing insurance claims of some 450 million Euro (ADAC, 2021). The German railway provider Deutsche Bahn expects asset damages of around 1.3 billion euros. Among other things, 180 level crossings, almost 40 signal boxes, over 1000 catenary and signal masts and 600 km of tracks were destroyed, as well as energy supply systems, elevators and lighting systems (MDR, 2021b). By April 2022, 11 of the 14 affected rail stretches are fully functional again. The less damaged stretches were functional again within 3 months, while some of the most damaged sections in the Ahr Valley are expected to be finished by the end of 2025 (DB, 2022). In Belgium, approximately 10 km of railway tracks and 3000 sleeper tracks have to be replaced, 50 km of catenary needs to be repaired and 70,000 tonnes of railway track bed needs to be placed, with estimated costs between 30-50 million Euro (Rozendaal, 2021a). Most damages have been repaired within two weeks. The most severely damaged railway line (between the villages of Spa and Pepinster) was reopened again on October 3, 2021 (Rozendaal, 2021b). In the Netherlands, no large-scale damage has been reported to transport infrastructure. A few national

highways were partly flooded (e.g., the A76 in both directions) or briefly closed (<3 days) because of the potential of flooding. Most likely due to relative low flow velocities, damage to Dutch national road infrastructure was limited. Several railway sections were closed (e.g., the railway section between Maastricht and Liege) and some damage occurred to the railway infrastructure, in particular to the electronic 'track circuit' devices and saturated railway embankments (Prorail, 2021).

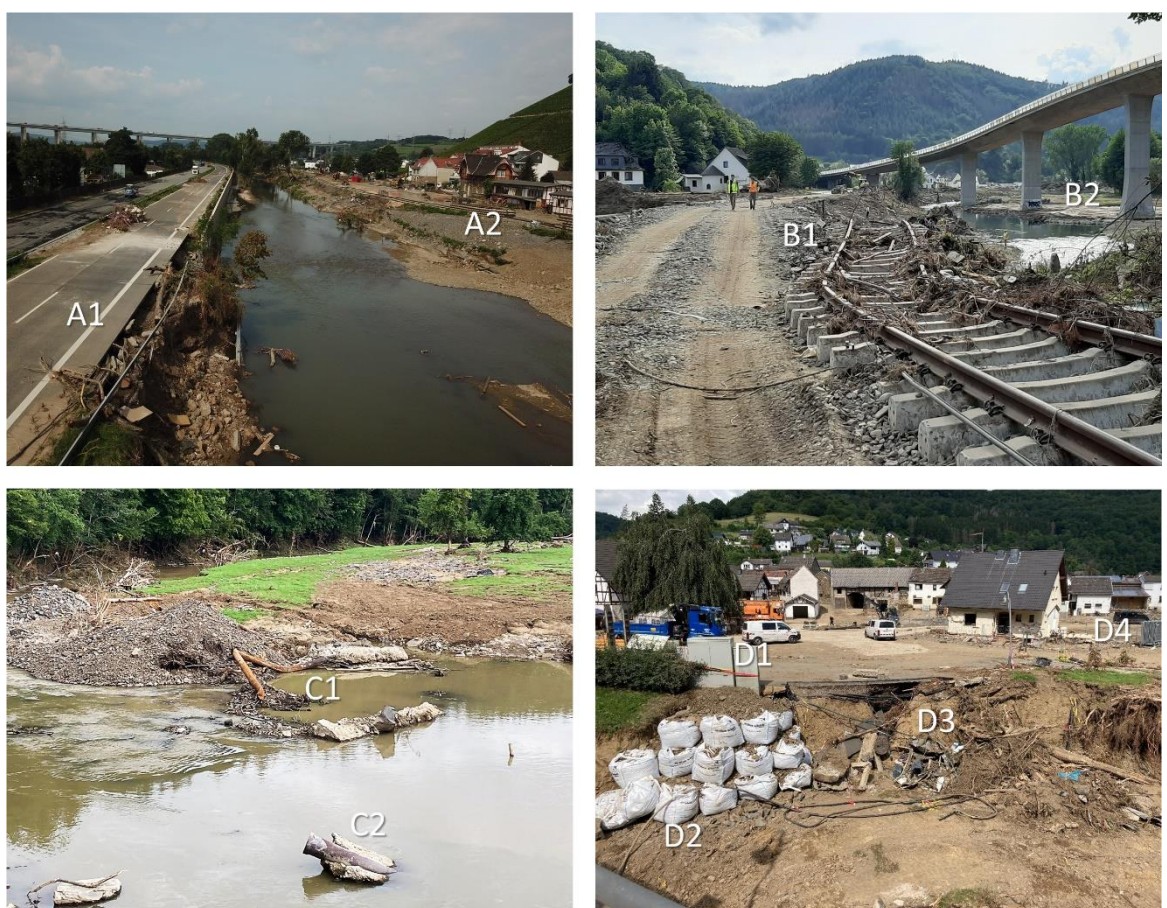

**Figure 1. Damage in the Ahr Valley, Germany (images taken on 11 August 2021). Top-left: destruction of federal highway B266 (A1) and railway (A2) near Heimersheim. Top-right: further upstream in the Ahr valley (Altenburg), large stretches of the Ahrtalbahn railway have been destroyed (B1) and the few remaining road and rail bridges show signs of temporary repairs (B2). Bottom-left: river bed erosion uncovered and destroyed many cables supposed to lie more than 80 cm below surface level (C1) as well as sewers (C2). Bottom-right: inundated electricity distribution infrastructure (D1), road erosion and stabilisation (D2), uncovered cables (D3) and collapsed buildings in Schuld. Pictures by Margreet van Marle/Deltares/GEERassociation, distributed under Creative Commons Attribution 4.0 license.**

## 2.2 Electricity and gas supply

At the peak of the event, around 200,000 people experienced power outages in Germany. Electricity infrastructure has been severely damaged in North Rhine-Westphalia and Rheinland-Pfalz. However, within two days around 50% of the power was restored through repairs and temporary fixes. Within eight weeks, no emergency power generators were required anymore,
with most of the power infrastructure restored in Germany's affected areas. Some areas, however, only had permanent power infrastructure after six months(Westnetz, 2022). The gas distribution network in the Ahr valley has been severely damaged. Approximately 133 km of natural gas pipelines, 8,500 gas metres, 3,400 house pressure regulators, 7,220 of the approximately 8,000 household connections and 31 gas pressure regulating and measuring systems have been damaged or destroyed (SWR, 2021). Gas supply was almost fully restored within 4.5 months after the flood event (Energienetze Mittelrhein, 2021). In
Belgium, approximately 41,500 people experienced power outages at the peak of the event. This was the result of both damaged and deliberately switched off electrical cabinets to prevent serious damages. It took around three weeks to fully restore power. Similar to Germany, severe damage has been observed to the gas network. In the villages around Liege, such as Chaudfontaine and Pepinster (Belgium), gas supply was fully recovered within five months (Grosjean, 2021). In the Netherlands, 1000-2000 households experienced a loss of electricity supply at the peak of the event. Between 100 to 200 households had no gas supply.
Within several days, electricity supply was restored (Task Force Fact Finding Hoogwater, 2021).

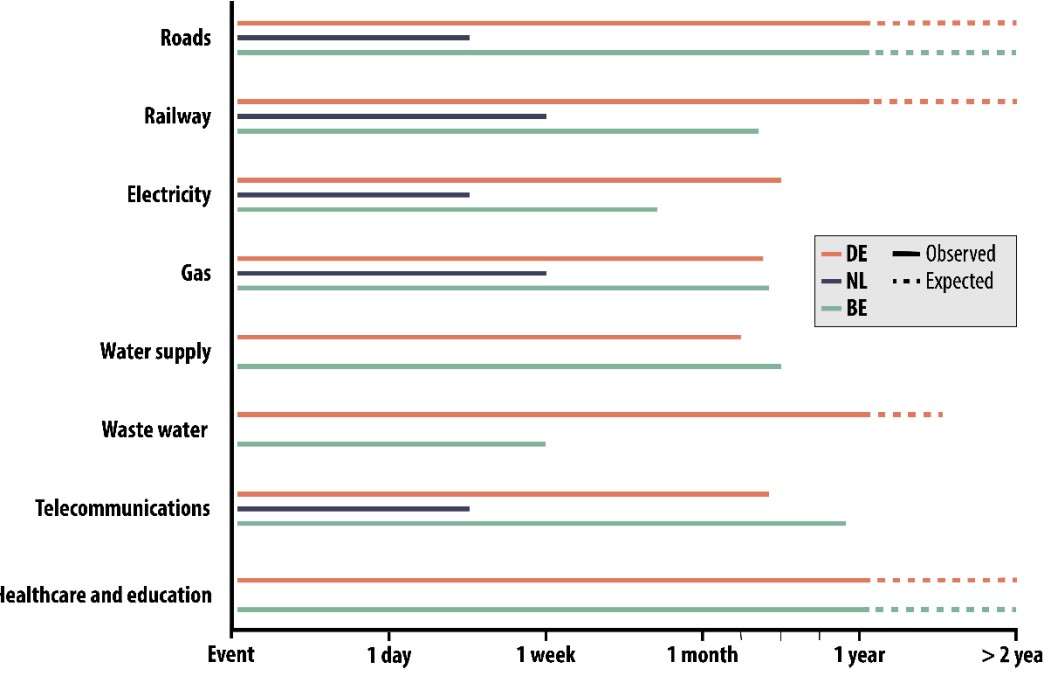

**Figure 2. Overview of observed and expected reconstruction duration of each infrastructure sector considered in this study. It should be noted that this figure presents reconstruction efforts of the system. No line indicates that no impacts are observed. Solid waste is**
95 **not included in the figure, as no impacts were recorded within each country.**

## 2.3 Drinking water supply and waste-water

In the region of Rheinland-Pfalz (Germany), most drinking water supply was restored within two months (Hochwasser Ahr, 2021a). However, sewage treatment plants in Altenahr, Mayschoss and Sinzig have been largely destroyed (Hochwasser Ahr, 2021b) and it is expected to take at least 1.5 years to fully repair most sewage treatment plants. Emergency sewage treatment plants have been built in the meantime (GA, 2021). In the Erft region 7 out of 31 wastewater facilities have been destroyed. Many facilities reported inundation of oil and diesel, forming layers up to 15 cm thick (Kuhn, 2021). In addition, much of the groundwater (and soil) in the flood region was mixed with oil (from destroyed residential oil tanks), chemicals such as fertilisers (from wineries and other agriculture) and chemicals from nearby industrial plants. In Sinzig, 3.6 million litres of oil-water mixture was recycled, gaining 3600 m3 of oil, to be reused for heating and industrial usage (Kuhn, 2021). n the heavily destroyed town of Bad Münstereifel (in the state of Northrhine Westphalia,), drinking water supply was reestablished within five days after the flood event (most frequently through emergency tanks), and about 50% of the city centre was re-connected to the fresh-water network shortly thereafter, however, water had to be boiled before consumption until about one month later (Bad Münstereifel, 2021). In Belgium, several towns experienced disruptions in water supply (in particular as a result of pollution). Directly after the event, approximately 3400 families had no access to potable water. Within less than a week, this was reduced to around 1650 families (Terzake, 2021). It took, however, six months to rebuild the permanent water supply infrastructure (SWDE, 2022). In The Netherlands, little to no problems have been recorded with regards to water supply.

## 2.4 Solid waste

We found no information regarding direct impact on solid waste facilities as a result of the flood event. However, there is a large pressure on the solid waste sector to clean the affected areas. One month after the event, we observed dozens of large temporary waste fills and frequent incidences of oil pollution in Rheinland-Pfalz during a field visit. In the Ahrweiler district alone, the flood caused as much solid waste as normally would be collected over thirty years. In Belgium, the amount of solid waste is estimated around 160,000 tonnes, stored at several places, such as the abandoned highway track A601. This highway has been used for approximately 9 months as a temporary storage for debris (Couplez, 2022). In the Netherlands, there have been primarily problems with waste deposits along the river banks, which is mostly the solid waste transported by the river from further upstream. Thousands of tons of tree debris (logs and deadwood) were recycled in the Ahr valley. For instance, the towns of Hoenningen and Mayschoss, served as major recyclic hubs. Approximately 500 tons of wood debris was transported, cut, chipped and recycled into firwood per day, following for at least 6 weeks post flood (Gather, 2021).

## 2.5 Telecommunication

In Germany, all severely affected areas experienced disruption of mobile network services. Within the region of Rheinland-Pfalz, it took two weeks to ensure 100% coverage again through emergency communication masts. Within one month, most of the network was restored to pre-disaster service provision. After five months, broadband has also been restored in the most affected areas, which started in most areas only after power infrastructure was rebuilt (Westnetz, 2021). In Belgium, it has

taken around 11 months to restore connection to the last communities within the affected area. In The Netherlands, approximately 7000 households were affected by disrupted services of telecommunication. This was primarily due to flooded telecommunication infrastructure in the direct vicinity of flooded houses. However, some distribution cabinets were flooded as well, with the largest flooded cabinet affecting around 700 households. Due to damaged bridges, several fibre cables were damaged. Five telecommunication masts were affected as well, but 'tuning' of the network ensured that the service disruption was kept to a minimum (Task Force Fact Finding Hoogwater, 2021).

### 2.6 Healthcare and education

In Germany, an estimated 180 general practitioner practices have been affected by the flood event. Impacts range from completely destroyed to unable to operate due to a lack of running water and electricity (Ärzte Zeitung, 2021). After 1.5 months, medical care was guaranteed again in the most affected regions in Rheinland-Pfalz (Hochwasser Ahr, 2021d). In the state of North Rhine-Westphalia, approximately 68 hospitals have been affected, of which several have been affected severely and will take at least 1.5 years to be rebuilt (Figure 2). Direct damages are estimated to be at least 100 million Euro to repair all medical facilities (Korzilius, 2021). In the town of Eschweiler (Germany), for example, the basement of the hospital was flooded, as well as the outbuildings and the entire outdoor area. The power supply collapsed, the entire building technology was destroyed and some 300 patients had to be evacuated by helicopter. Property damage is expected to be around 50 million Euro. Within 3.5 weeks the hospital was partly operational and within three months, all hospital operations continued normally (SAH Eschweiler, 2021). The Mutterhaus Ehrang hospital in Trier (Germany) is now permanently closed as the hospital is too severely damaged to rebuild. Furthermore, in the region of Rheinland-Pfalz (Germany), 19 day-care centres and 17 schools suffered damage from the floods, affecting more than 8000 students (Staib, 2021). Approximately four months after the flood event, the district of Bad-Neuenahr Ahrweiler established emergency educational facilities using 297 containers that serve as classrooms, offices, and dining facilities for more than 800 students (Wiesbadener Kurier, 2021). In Belgium, various rural clinics have been affected and were unable to provide any services. Concurrently, in the most affected areas, general practitioner facilities have been completely destroyed (Le Spécialiste, 2021). In the Netherlands, one nursing home was flooded, and one hospital was evacuated as a precautionary measure.

### 3. Have research studies already "reflected" such impacts?

Most often, large-scale object-based infrastructure impact studies (e.g. Bubeck et al. 2021) only disclose aggregated risk metrics (i.e. country-level risk estimates), which hampers verification and validation with observed impacts on smaller scales. Van Ginkel et al. (2021) assessed river flood risk for all road segments in Europe. Of the eight motorway floods incidents in Germany reported by Hauser (2021), three are recognizable as flood hotspots in Van Ginkel et al. (2021). During the event in 2021, most damage was caused by relatively small rivers which are only represented in the hazard data from the point that the upstream catchment is above 500 km$^2$. For example, the Ahr Valley is partly covered (400 of the 900 km$^2$) by Van Ginkel et al., who estimate the road repair costs at 4 to 29 million euro (under low and high flow velocities resp.) for a 1:500 year event.

The field visit showed damage caused by high flow velocities at multiple places, and video footage of the events suggests these are locally more towards 2 m/s, which van Ginkel et al. considered 'high flow velocity', than towards 0.2 m/s, which they considered 'low flow velocity'. At first sight, the spatial extent of the exposed assets has reasonable correspondence to the model of Van Ginkel et al. (2021). However, the model ignores bridge damage, which in reality was a major source of damage (Section 2.1). Also, a significant share of observed damage resulted from pluvial flooding, flash flooding, and landslides which

was not captured by Van Ginkel et al. (2021).

    Reconnaissance observations (August 2021) along the rivers Ahr and Erft (Lemnitzer et al., 2022) documented severe, as well as irreparable damage to bridges designed and constructed within the last two decades; and total destruction of almost all historical bridges, typically constructed on shallow foundations. Historical bridge designs concentrated primarily on cross-sectional requirements for expected water volumes. Triggered by flood events in the past four decades, bridge design research

has broadened by focusing on risk-based scour assessment, hydrodynamic pier designs, reduction of intermediate bridge support elements, impact and collision loading, implementation of bridge protection mechanisms such as from wood debris, as well as machine learning approaches from past failures (VAW 188, 2006, Bento et al., 2020, Majtan et al., 2021, Naser, 2021). Accounting for all these mechanisms, however, is complex (Haehnel and Daly, 2002) and no guarantee to avoid the observed failures. Various international design codes (e.g., American Bridge Standard AASHTO, Australian Bridge Standard

AS5100, and Japanese Bridge Standard SHB) provide quantitative tools to assess impact loading from debris/ tree logs, however, bridges erected prior to recent design requirements are unable to maintain global structural stability under the excessive multidirectional loading, such as seen in the 2021 floods.  Based on field observations, the advancement of erosion prevention practices for flood events emerged as a critical research focus, as the interface stability between water, soil and foundation elements was found to be compromised at almost all bridge damage locations visited.

Next to the above insights into modelling direct physical damages, we can also use the observation from the event to further improve and validate our assumptions on post-disaster (infrastructure) recovery. In particular when modelling the economic and societal impacts, the recovery process is one of the most important drivers of losses (e.g., Koks et al., 2015). The July 2021 event has taught us several insights. Firstly, there is a prioritisation between different infrastructure systems. In Germany, for example, we found that in several affected areas the electricity network was repaired first, which was subsequently followed

by the gas and broadband network. Secondly, there is a prioritisation within infrastructure systems. For example, more critical roads are repaired sooner than less critical roads. While this may sound obvious, academic studies often consider a recovery of the entire system, without considering a specific order of importance. Finally, good recovery management practices and enough trades(wo)men (e.g. electricians, utility workers) are one of the most important drivers of a speedy and successful recovery. These are often not included within the modelling assumptions.

## 4. Moving Forward

Based on our findings, we highlight three aspects to move forward in the field of infrastructure disaster risk assessments. First, merely focusing on flood extent and depth is not sufficient to estimate the impacts of extreme flood events to infrastructure. In particular in Germany and Belgium, it became evident that the high flow velocities (resulting from the local topography and the intensity of the rainfall) are a decisive factor in explaining the degree of destruction. Many of the observed failures such as bridge scour, road embankment instabilities, and erosion of aggregate foundations could likely better be explained from flow velocity rather than flood depth. Future flood impact studies, especially those focusing on transport infrastructure, should aim to account for flow velocity in their impact modelling. In particular in areas with steep gradients.

Second, the observed impacts on CI highlight the influence of spatial scale on the magnitude of the impacts. On a local and regional level, the disruptions in daily lives and to the economy were enormous. Yet, zoomed out on a national scale, the impacts were *relatively* small. While large-scale studies are useful to identify potential hotspots and bottlenecks in the system, local-scale studies are essential to better understand the real impacts (and are also better able to do so). This is true for both the consequences to infrastructure assets and the services, and the impacts on lives and livelihoods.

Finally, the level of destruction and disruption caused by this event highlights the need for the development of both asset and system-level adaptation measures, securing more resilient infrastructure systems. Extreme weather events are expected to become more likely in Western Europe, but also globally in an increasingly warmer world. As such, there is an urgency to not only investigate how service provision can be ensured in the case of an extreme event, but also how the recovery process to a minimum service level can be as swift and smooth as possible. This calls for a further collaboration between the different sectors of reliability and systems engineering, and disaster risk modelling and management. The limited number of studies on impact on CI due to flooding highlights the need for more detailed infrastructure failure impact assessments including cascading impacts.

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
