# Peer review of "Brief Communication: Critical Infrastructure impacts of the 2021 mid-July western European flood event"

_Natural Hazards and Earth System Sciences, 2021_

## Author Comment (AC1)

**Response to Alexander Fekete**

Overall, the paper addressed an important aspect of these flood events; critical infrastructure failures. It is helpful for research to have such an account of CI interruption and recovery times in a paper. The paper presents findings based on a mix of research and newspaper sources, and own field observations. Three countries are covered, but the reports and findings are mostly on Germany. The overall structure could be improved a bit, but the overall documentation of damages is helpful for other researchers.

*We thank Prof Fekete for his kind words and acknowledge that the structure could be improved. His suggestions will ensure that it will.*

Content comments:

- Page 4, line 85. Gas supply is almost fully restored. Yes, from the perspective of the utility providers. But many homes are still disconnected. There is a shortage of heating installations for the houses. But I guess, if this is what you mean by "almost", it is ok as it is.

*This is a good point, we do mention in the introduction that we focus here on the "large-scale" infrastructure systems, and that we do not look into individual connections to houses, as that is almost impossible to find and is very case specific.*

- 2.3 Lines 99-104: it is correct, but maybe you need to add that in most places, emergency sewage tanks have been installed, by THW and others.

*Thank you, this will be added to this section.*

- Line 106: add a source, if you have

*This reference accidentally got removed. We did another thorough check and an update of references. Everything should be correct now (some other refs were also missing)*

- 2.4 This section is quite short, by comparison. Since there is no conceptual order of critical infrastructure, and you also do not cover many more CI sectors/ branches; you may consider shifting this information in 2.4 to another section that summarizes 'other impacts'.

*We have actually been able to provide some more information thanks to some of the other reviewer's suggestions. Hence, we decided to keep this section.*

- Section 2 in general: I would suggest either having at least a sentence on each of the three countries in each subsection (2.1 etc). Or, omitting short sentences such as "In The Netherlands, little to no problems have been recorded with regards to water supply." However, especially Belgium is not mentioned in every sub-section yet.

*Agreed, all countries should (and will be) mentioned in all paragraphs, also when no information or impact is found (but then kept very brief that we simply state that nothing happened and/or no information is available).*

- 2.6. For Germany, more hospitals heavily affected are missing (Erftstadt, Leverkusen). The paper is mixing overview information such as 105 general practitioner practices with one more elaborate example of one hospital; maybe you could add more overview information (x numbers of hospitals) as well.

*Thank you for this suggestion, we have been able to find quite some extra information on hospitals. Therefore, we have rewritten Section 2.6 to accommodate this.*

- Section 3 has the same heading as section 2. And also the content appears to be the same.

*It seems something went wrong when we put everything in the right NHESS Format. The section 3 title will be updated.*

- So maybe it is better to merge both sections. Section 3 appears very much like miscellaneous observations, the 2.7 'other' However, the style is even more narrative and overall, it is not so clear, which fields the authors wish to cover and which not. And again, mainly Germany is covered. Maybe this section could be divided into

  2.7 Roads (first para)

  2.8 Bridges (third para)

  And the middle para on shortages in flood zonation mapping be shifted to section 4?
  Or, a new 2.9 on retention basins as infrastructure overlooked so far? The information is fine and it is also fine that the authors cite themselves, it seems. However, maybe reduce mentioning the name of one author a bit; make it more elegant by replacing it by "the same study" or similar.

*Section 3 is meant to compare some of the impacts with academic findings, how this overlaps and how the empirical findings can provide guidance on moving forward in the academic field. Hence, we keep them separate (but with an updated title). We have also now included another paragraph on lessons learned for recovery modeling.*

- You could add similar studies on road and rail interruption studies such as by Atzl/Keller. But the paragraph itself is very important and very laudable that the authors observe further needs of assessment in (their) previous studies as well. And indeed, flood assessments now need many updates including bridges collapsing, sediment transport, tree transport and many more.

*As we are very tight on space, we have decided not to include a comparison with studies by Atzl and Keller. They have also primarily focused on describing past events, while we try to make the link to damage modeling in this particular section of the manuscript.*

- Lines 151-154 Interesting argumentation of the "counterfactual", but I agree that the static maps of riverine flood zonation do not match (every) real event. Since they are based on scenarios that see all potentially flooded areas, irrespective of the flood waves (and duration of discharge). So if the message here is that the study was counterfactual because the mining pit 'saved' other areas from being

flooded, which was not captured by the study in 2020, or by the flood maps, it is correct and fine.

*That is indeed our message, but we have actually decided to remove this paragraph and replace this with a paragraph focused on recovery, to broaden the scope of this section, while staying within the word limit.*

- The three findings are not surprising and sound a bit generic, given the existing literature in science about this. But maybe it is still very important to stress these points once more after this important event, so as to hope to improve the risk management practices in Germany, at least.

*Yes, we agree the conclusions may not be very exciting, but we kind of felt we need to reiterate these points as we are still not there yet in most risk studies.*

**Response to Michel Journee**

*We thank Mr Journee for this extremely valuable comment. We have, however, decided to remove the figure, as it has proven to be difficult to make a proper figure that aligns all the different country-level information. As other studies focus much more on the meteorological and hydrological conditions of this event, we will simply refer to those studies (e.g. Mohr et al., 2022).*

**Response to Knieps:**

Information to 2.4 Solid waste:

Destroyed and silted up furniture and household appliances: On the flood night in July, as much bulky waste has accumulated in the Ahrweiler district as it would otherwise in 30 years. The district administration spoke on Thursday of around 240,000 tons of flood bulky waste so far. In the future, around 10,000 tons a day would also be transported away from interim storage facilities in the Ahr Valley.

The waste management company of the Ahrweiler district (AWB) reckons with at least another 50,000 to 100,000 tons of bulky waste from the disaster area.

Source: https://www.sueddeutsche.de/panorama/hochwasser-bad-neuenahr-ahrweiler-ahrweiler-durch-flut-sperrmuell-wie-sonst-in-30-jahren-dpa.urn-newsml-dpa-com-20090101-210902-99-67800

*We thank the reviewer for providing some input for section 2.4. This will be integrated in the manuscript in section 2.4.*

**Response to Anonymous Ref 2:**

This brief communication provides an overview of the impact of the July 2021 flood (Germany, Belgium, the Netherlands) on important (critical) infrastructure systems (Transport Infrastructure, Electricity and gas supply, Drinking water supply and waste-water, Solid waste, Telecommunication, Healthcare and education) as well as the progress in rebuilding these important systems. Problems in the field of risk assessment

of infrastructure disasters are identified and statements are made for future activities to improve risk modeling for critical infrastructure. Thus, the draft meets the content requirements for a brief communication. Overall, the manuscript needs some revisions; however, the summary of information from media and other work is an important contribution to research on the highly topical example of a severe extreme event that was not foreseen and has highlighted our limitations in risk management.

**Minor major comment:**

- The authors should rethink the structure in Section 2 and 3.

*As we have also indicated to another reviewer, section 3 is meant to compare some of the impacts with academic findings, how this overlaps and how the empirical findings can provide guidance on moving forward in the academic field. Hence, we keep them separate (but with an updated title).*

- The argumentation of the paragraph L149-154 is interesting, but does not quite fit very well.

*We believe it is an interesting comparison, as the study by Fekete (2020) is one of the few studies that looked at the affected area. However, we have actually decided to remove this paragraph and replace this with a paragraph focused on recovery modeling, to broaden the scope of this section, while staying within the word limit, also based on the reviewer's next comment.*

- Section 3 "Critical Infrastructure Impacts" really only addresses the transportation infrastructure aspect; can you please comment a bit more on the other topics discussed in section 2?

*We agree that this is an important point. We will add a paragraph in which we discuss the lessons learned for recovery modeling.*

**Minor comments:**

- The following specifications for "Brief communications" from the publisher side were not met:

  https://www.natural-hazards-and-earth-system-sciences.net/about/manuscript_types.html
  „Brief communications have a … maximum 20 references, and an abstract length not exceeding 100 words." Even if I find the aspect with the citations (caused by various media reports) less relevant and it is rather important to link/appreciate the work of others --> Decision of the editor/publisher

*Yes we agree that we have exceeded the number of references, but it's hard not to exceed that in our case. We should link to the news articles in which we have found the information.*

- L19: See also for Germany:
- Junghänel, P. Bissolli, J. Daßler, R. Fleckenstein, F. Imbery, W. Janssen, F. Kaspar, K. Lengfeld, T. Leppelt, M. Rauthe, A. Rauthe-Schöch, M. Rocek, E. Walawender u. E. Weigl (2021): Hydro-klimatologische Einordnung der Stark- und

Dauerniederschläge in Teilen Deutschlands im Zusammenhang mit dem Tiefdruckgebiet „Bernd" vom 12. bis 19. Juli 2021, Deutscher Wetterdienst 2021 https://www.dwd.de/DE/leistungen/besondereereignisse/niederschlag/20210721_ bericht_starkniederschlaege_tief_bernd.pdf

- Maybe there are similar reports from the other two weather services for the respective country?

- Figure 1:

- A detailed integration of the relevant rivers in all three countries would be very desirable; it is unclear which data come from which dataset; How (and by whom) exactly was the flooded area determined? When is this flooded area evaluation from? Country boundaries could be a bit thicker for better distinguishability? The city names should be in English (e.g. Köln à Cologne); Why is the scale 200 - 225 mm used in the colorbar, when it is not used in the figure? This gives a false impression of the maximum values.

*We have decided to remove the figure, as it has proven to be difficult to make a proper figure that aligns all the different country-level information. As other studies focus much more on the meteorological and hydrological conditions of this event, we will simply refer to those studies (e.g. Mohr et al., 2022).*

- L20: „Rhine" In Germany, the tributaries such as the "Ahr" and "Erft" were particularly affected.

- L23ff: Newly published "Press release from Munich RE from 2022/01/10": https://www.munichre.com/en/company/media-relations/media-information-and-corporate-news/media-information/2022/natural-disaster-losses-2021.html

  How are the loss amounts to be understood? Total loss or Insured loss; Direct loss or Indirect loss?

*Good point, we will rewrite these sentences with the latest insured loss estimates that we could find.*

- The captions of Section 2 and Section 3 are identical.

*This accidentally happened when converting to the NHESS format. This will be fixed in the new version.*

- Figure 2: To avoid confusion; the damage was not on 11 August 2021, but the images are from the 11 August --> Recording date

*Thanks for raising this, this has been fixed. This is now Figure 1, and the date of the pictures taken is indicated in the caption.*

- Figure 3 is not referenced in the text.

*Figures have been updated and are now referenced properly in the text. Thank you for the reminder.*

- L140 „…which hampers verification and validation with observed impacts." Suggestion: Maybe add „on smaller scales"

*We agree, this has been added to the text per reviewer recommendation. (see revised text in Section 3)*

- L140ff: I miss in the reference the detailed literature citation from "Van Ginkel et al. (2021)". On which data/models are the estimations of the river flood risk based for all road segments in Europe?

*The reference is now added. In Van Ginkel et al. (2021) we have used LISFLOOD and OpenStreetMap data.*

- L141: "…for all road segments in Europe." Really for all? Or only for a part of the "road categories" (e.g. highways, trunk roads,...)

*Yes, for all road segments that are mapped as 'road' in OSM, which is nearly complete. This includes 'motorways', 'trunk roads', 'primary', 'secondary', 'tertiary' and 'other', the latter including the remaining underlying smaller road types such as residential roads and tracks and trails. See for more details: https://nhess.copernicus.org/articles/21/1011/2021/.*

- L143: „… relatively small rivers…" Please give examples

*Thank you for this question, this is important to clarify. In the hazard data (LISFLOOD model), all upstream rivers/creeks are ignored from the point that their upstream catchment is < 500 km². We have added this to the text, and clarified for the Ahr River example.*

- L143: „During the event, …" Suggestion: „During the event in 2021, …"

*We agree with this suggestion, the text has been changed per reviewer suggestion.*

- L143: I miss in the reference the detailed literature citation from "Dottori et al., 2021". As this is already the second missing citation; please check again if all citations are mentioned in the references.

*We agree that this has been sloppy from our side. We have double checked everything and all citations are now included in the new version.*

- L145: "…flow velocities at multiple places." Could you provide values here and make an assessment of them?

*We did not measure or reconstruct the flow velocities during the fieldwork. However, from a quick look at the video footage of the event and a comparison with the thresholds set in Van Ginkel et al., we could conclude that locally the flow velocities have been much more towards Van Ginkel et al. 's 'high flow' (2 m/s) than 'low flow' (0.2 m/s).*

- L146: „…correspondence to the model". Unclear what exactly is meant; to the model in Van Ginkel et al.?

*We agree that this has been unclear, this text has been modified in section three. The reference is now added and additional text has been provided.*

- L147: "Figure 2". Maybe better suitable: Reference to Section 2.1

*We agree that we should reference to section 2.1.*

- L157f: "…freeboard requirements." Please explain.

*Thank you for this request. Yes, this might have not been clear. We refer to the clearance (i.e., cross-sectional volume/area) available between the water level and bridge superstructure/deck in case of an event to accommodate the increased water volume from a cross-sectional perspective. The sentence in Section 3 has been rephrased to avoid confusion and to convey the intent of the statement better.*

- L171ff: Another problem here is that the data basis is miserable.

*We agree with the reviewer and modified the text accordingly.*

- Section 4:Comment from RC1: "The three findings are not surprising and sound a bit generic, given the existing literature in science about this. But maybe it is still very important to stress these points once more after this important event, so as to hope to improve the risk management practices in Germany, at least." I agree with this statement

*Yes, we agree the conclusions may not be very exciting, but we kind of felt we need to reiterate these points as we are still not there yet in most risk studies.*

**Response to Anonymous Ref 3:**

The authors present a "Brief Communication" on "Critical Infrastructure impacts of the 2021 mid-July western European flood event". The collection of information is helpful, but only limited scientific conclusions are presented. However, presenting the material as a brief communication seems justified to me. I agree with the authors, that the material can be helpful for further studies. I also agree with the comments of the other reviewers so far. I recommend publication of the paper after some minor corrections.

I only have a few additional remarks:

- Figure 1: More detailed rainfall maps for Germany and neighboring countries are available in this report:

  https://www.dwd.de/DE/leistungen/besondereereignisse/niederschlag/20210721_bericht_starkniederschlaege_tief_bernd.pdf?__blob=publicationFile&v=10 (e.g. Figure 6).

*We have decided to remove the figure, as it has proven to be difficult to make a proper figure that aligns all the different country-level information. As other studies focus much more on the meteorological and hydrological conditions of this event, we will simply refer to those studies (e.g. Mohr et al., 2022).*

- Section 2.4: Additional information on solid waste for the Ahrweiler region is available on the website of the 'Kreisverwaltung':

[https://kreis-ahrweiler.de/es-waren-keine-abfaelle-sondern-erinnerungsstuecke/](https://kreis-ahrweiler.de/es-waren-keine-abfaelle-sondern-erinnerungsstuecke/)

*We thank the reviewer for this additional information, this has been included now.*

- Line 143: Dottori et al. is missing in the reference list.

*We thank the reviewer for pointing this out, reference was added*

**Minor editorial remarks:**

- Line 60: „euro" -> „Euro"; „40km" -> „40 km"
- Line 129: „euro" -> „Euro"

*We thank the reviewer for these minor editorial remarks, this has been fixed accordingly.*

---

## Author Response (AR2)

**Final Response**

We thank the editor and the reviewer for their positive words. Please find below a response to the few final small corrections requested by Reviewer #2:

• *LI21: Citation is missing in the references*

**Citation has been added**

• *LI29: I can't find any reference of GDV of 2022 in the references. Have you updated it? There is one from 2021, which is from December 2021, but that is certainly not the "latest" one… (e.g., https://www.gdv.de/gdv/themen/schaden-unfall/naturgefahrenreport-2022-klimaresiliente-kommunen-105824)*

**Both the insured loss estimate and the citation have been updated to the most recent one.**

• *L104: m3 --> m³*

**This has been fixed**

• *L104: Missing "I" at the beginning of the sentence*

**This has been fixed**

• *L193: "flow velocities"*

also streamflow (in m³/s)

**The sentence has been rewritten as follows:**

..it became evident that the high discharge and streamflow, and corresponding high flow velocities..